# Dental Transmigration: An Observational Retrospective Study OF52 Mandibular Canines

**DOI:** 10.3390/biology11121751

**Published:** 2022-11-30

**Authors:** César Martínez-Rodríguez, Natalia Martínez-Rodríguez, José María Alamán-Fernández, Pedro Luis Ruiz-Sáenz, Juan Santos-Marino, José María Martínez-González, Cristina Barona-Dorado

**Affiliations:** 1Department of Dental Clinical Specialties, Faculty of Dentistry, Complutense University of Madrid, 28040 Madrid, Spain; 2Surgical and Implant Therapies in the Oral Cavity Research Group, Complutense University, 28040 Madrid, Spain; 3Department of Dentistry, Central Hospital of the Red Cross of Madrid, 28003 Madrid, Spain; 4Department of Surgery, Faculty of Medicine, University of Salamanca, 37007 Salamanca, Spain

**Keywords:** mandibular canines, transmigration, panoramic radiography, cone beam computed tomography (CBCT), clinical implications, radiological characteristics

## Abstract

**Simple Summary:**

Transmigrated canines represent a rare eruptive entity that usually occurs at the mandible. The diagnosis is usually made after radiographic examinations, such as panoramic radiography. Its observation should be analyzed in detail with computerized scanner studies of the CBCT type. Although, in most cases, they are usually asymptomatic, the appearance of odontomas or the risk of developing tooth cysts should be considered. The age of diagnosis and its position will be decisive to assess dental replacement by orthodontic treatment. When this is not possible, surgical treatment is advised.

**Abstract:**

The aim of this study was to analyze the prevalence of transmigrated canines in a Spanish population by evaluating their clinical and radiological characteristics. The descriptive observational study obtained 6840 orthopantomographs from all patients seeking dental care in the years 2017–21 via the Patient Reception Service and Dentistry Service at the Faculty of Dentistry at the Complutense University of Madrid and the Central Hospital of the Red Cross of Madrid (Spain). In total, 52 patients presented transmigrated canines, establishing a prevalence of 0.76%. This sample comprised 28 women and 24 men. Whenever a transmigrated canine was identified, a CBCT scan was obtained and used to evaluate the clinical and radiological variables associated with canine transmigration. The predominant side of the transmigration was the left (57.69%) compared to the right side (42.30%). The position of the canine, in order of frequency, according to the Mupparapu classification, corresponded to type IV (42.30%), type II (36.53%), type I (15.38%), and type V (5.76%), with no type III transmigrations found. Clinical manifestations were only recorded in 17.30% of cases, and 11.53% of the radiological findings showed the presence of tooth cysts that were confirmed by histopathological studies. Other impactions, in addition to the transmigrated canine, were found in five patients (9.61%), with the majority being the presence of third molars.

## 1. Introduction

The phylogenetic evolution of the human species has resulted in, among other things, a reduction in the size of the maxilla and mandible; thus, the inability of part of the dentition to erupt is a frequent finding. This situation is called “dental inclusion” [1].

After the third molar, the canine is the second most commonly impacted tooth, with a frequency of between 0.2% and 3.6% in the upper maxilla and between 0.35% and 1.29% in the mandible [2,3].

Occasionally, these impactions occur at some distance from the eruption site, in which case they are described by the term ‘ectopic eruption’. There are various types of ectopic disorders, including the phenomenon of dental transmigration [4,5,6].

The term transmigration was first introduced by Ando in 1964 to describe the phenomenon of physiological migration. In 1971, Tarsitano et al. [7] put forward a definition of transmigration as a phenomenon in which the unerupted canine crosses the dental midline. In 1985, Javid [8] introduced a modification to this concept, defining transmigration as a situation whereby over half of the non-erupted impacted tooth length is forced to cross the midline.

Research has investigated and subsequently rejected several different factors that could potentially explain transmigration, and its precise etiology remains unclear [1].

At the same time, there would appear to be an association between transmigration and certain pathological processes such as follicular cysts, odontomas [9], supernumerary teeth, the premature removal of a temporary canine [10], enamel hypoplasia, or class II malocclusion with a deep bite, which could be the cause of the transmigration [11,12,13].

In 2002, Mupparapu [14] proposed a system for classifying the intraosseous transmigration and ectopic eruption of mandibular canines according to their migration pattern and position in the mandible in relation to the dental midline, identifying five types (Figure 1):

Type I: canine positioned mesioangularly across the midline, labial or lingual to the anterior teeth.

Type II: canine horizontally impacted near the inferior border of the mandible, inferior to the apex of the incisor teeth.

Type III: canine erupting on the contralateral side.

Type IV: canine horizontally impacted near the inferior border of the mandible, below the apex of the posterior teeth on the contralateral side.

Type V: the canine positioned vertically in the midline, with the long axis of the tooth crossing the midline.

Transmigrated canines are generally asymptomatic, so diagnosis tends to occur casually during a routine radiological check-up. However, in some cases, transmigration may be accompanied by a clinical manifestation, such as the presence of pain resulting from inflammation or infection, or sensitivity disorders due to impaction [15,16].

According to the literature, transmigration is understood to be a very uncommon eruption disorder, and so its rarity has made it difficult to establish its prevalence among the general population, as most of the cases documented are isolated cases of wide heterogeneity.

The strength of this research work is given by the contribution of 52 new cases, with the aim to determine the prevalence of transmigration in a Spanish population and evaluate possible clinical and radiological characteristics and the possible pathological processes associated with transmigration.

## 2. Materials and Methods

### 2.1. Patient Selection

This multi-center, transversal, retrospective, observational study followed guidelines established by the Declaration of Helsinki (17) for research involving humans; the study design was approved by the Research Ethics Committee of the San Carlos Hospital (17/050-E), Madrid (Spain). All patients who attended the Patient Reception Service Dentistry Service, seeking dental treatment in the years 2017–2021 at the Faculty of Dentistry at the Complutense University of Madrid and the Central Hospital of the Red Cross of Madrid (Spain) were supplied with information about the design, characteristics, and purpose of the study and were asked to provide their informed consent to take part.

Panoramic radiographs (ORTOPHOS—Sirona Dental Systems GmbH; Bensheim, Germany) were taken of all patients (6480 patients) and assessed individually. In 52 cases, the presence of transmigrated canines was observed, and these patients underwent a Cone Beam Computerized Tomography (CBCT) scan (NewtomVGi, QR srl. Verona, Italy).

### 2.2. Evaluation of Clinical Data

An anamnesis was created for each patient presenting a transmigrated canine registering age, gender, and race.

Any clinical aspects were noted, such as the presence of a temporary canine, skeletal class, midline deviation, and adjacent teeth vitality, by means of a digital pulpometer (DP 2000. Denluxe), and the patients were asked why they had come to the clinic. They were thus classified into one of two groups: asymptomatic (patients attending for a routine check-up, whereby the discovery of a transmigrated canine was unexpected) or symptomatic (patients presenting symptoms such as pain, inflammation, infections, or sensitivity disorders).

### 2.3. Evaluation of Radiological Data

The following variables were registered in the radiological examination:

. Side: the direction (left or right) of the transmigration.

. Transmigration type: the type was classified according to Mupparapu’s classification (I, II, III, IV, and V).

. Rhizolysis of teeth adjacent to the transmigration.

. Other impactions: the presence of all impactions was registered, and both the number and the affected tooth or teeth were noted.

. Follicular enlargement (Figure 2): the size of the follicular sac was measured in millimeters using the NNT Viewer, establishing intervals of 0–3 mm and >3 mm [17].

### 2.4. Statistical Analysis

Statistical analysis was carried out using the SPSS 22.0 statistical software package (SPSS Inc, Chicago, IL, USA). A descriptive analysis of the quantitative and qualitative variables was performed, calculating the means, standard deviations, ranges, and frequencies. The chi-squared test was applied to determine the influence of the position variable. Pearson’s correlation coefficient was used to determine whether there was a relation between the position and the other parameters. The statistical significance was established as *p* < 0.05 in all the statistical tests.

## 3. Results

Of the 6840 patients who were seen by both centers, only 52 patients presented with the presence of a transmigrated canine, so the prevalence was 0.76%. The sample comprised 28 females (55.84%) and 24 males (46.15%), with a mean age of 34.91 years (SD: 12.52), ranging between 21 and 72 years. All patients were Caucasian.

The presence of a temporary canine was found in 48 cases (92.30%), with no deviation from the midline, as well as positive vitality in all the teeth adjacent to the transmigration.

The skeletal class was I in 49 patients (94.23%), 2 patients were class II (3.84%), and 1 patient was class III (1.92%).

The most common side of the transmigration was the left side, with 57.69% of the cases, compared to 42.30% on the right side, and no case of bilateral transmigration was found. According to the Mupparapu classification, the most frequent type, in decreasing order, was type IV (42.30%), type II (36.53%), type I (15.38%), and type V (5.76%). Type III transmigrations were not found (Figure 3 and Figure 4).

Clinical symptoms were observed in nine patients (17.30%), with the following distribution: six patients presented with sensitivity disorders corresponding to two cases of type II transmigration and another four of type IV (Figure 5), and three patients with episodes of swelling, with two belonging to type I and one to type II.

Follicular enlargement was observed in all patients, presenting the highest mean (2.75 mm) type I transmigrated canines, followed by type IV (2.06 mm). Additionally, six patients (11.53%) presented images compatible with cysts, which, after their removal and histopathological analysis, were identified as dentigerous cysts, being distributed first-mind in type IV.

The presence of other impactions, in addition to the transmigrated canine, was found in five patients (9.61%), with five patients presenting third molars and one patient presenting upper canines.

As for the treatment chosen to resolve the transmigrated canines, 29 patients (55.76%) underwent surgical removal, while the remaining 23 opted to remain under observation by means of regular clinical and radiological check-ups.

When transmigrated canine type (classification) was analyzed in relation to the variables age, gender, clinical symptoms, and the presence of cysts and other impactions, no statistically significant relations were identified (*p* ≤ 0.05 Pearson correlation) (Table 1).

## 4. Discussion

Transmigration, or, in other words, the migration of impacted teeth crossing the dental midline, is a rare phenomenon that has classically been reported as affecting mandibular canines, with transmigrated maxillary canines being even rarer.

The rarity of transmigrated upper canines may be due to the small space between the roots of the upper incisors, as well as a series of obstacles such as the floor of the nasal fossae, the maxillary sinus, or the mid-palatal suture that means that maxillary canines would require great force in order to migrate past them [18,19].

Most studies reported more transmigrated canine cases in females [3,9,20]; however, in the present study, the distribution by gender was not so evident, given that transmigration in females was 53.84%, with 47.26% in men. The largest series was obtained by Plakwicz et al. [13], which found 93 cases of transmigration, and 68.81% of patients were female.

The age of diagnosis cannot be considered as a determining factor since it influences the population studied. The age range in the present sample was 21–72 years, relatively older than most other studies with an orthodontic population, which have reported cases in patients as young as 7 years of age [10,13,21].

The exact etiological mechanism causing transmigrated canines remains unclear, and various theories have been put forward to explain this phenomenon. The loss of temporary teeth and occupation of the resulting space by adjacent teeth have been suggested by Azeem et al. [10]; however, in this study, in 92.30% of cases, the temporary canine was present. In the same line, authors such as Dalessandri et al. [9], Torres-Lagares et al. [22], Madiraju et al. [23], and Erdur et al. [24] have suggested that transmigrated canines may be accompanied by the presence of odontomas, although this association would not explain most cases of transmigration.

Finally, Plakwicz et al. [13] analyzed the largest series presented so far; they considered that class II malocclusion with a deep bite could be the cause of transmigration. Nevertheless, in this study, this class II malocclusion was only present in 3.84% of cases.

Mupparapu [14] established a classification system for transmigrated canines according to their position in the mandible, which has been widely used in the literature. In the present study, the most common type observed was type IV (42.30%), followed by type II (36.53%) and type I (15.38%). This low percentage of type I contrasts with the findings observed by Azeem et al. [10], who, in 25 cases, found that 76% of patients were type I. In the same line, findings were obtained by Plakwicz et al. [13], who found especially type I (64.5%), followed by type II (23.7%).

However, other authors such as Murat et al. [25], who in their review found a predominance of type II (47.05%), followed by type I (23.52%), as well as Herrera-Atoche et al. [26], for whom the highest percentage was type II (34.93%), followed by type I (31.6%).

Transmigration usually occurs without any noticeable clinical manifestation, although it may sometimes be accompanied by symptoms of adjacent teeth [18]. The first cases were published by Caldwell [27] and Bruszt [28] as a result of neurological disorders produced by the compression of the inferior alveolar nerve by the impacted tooth. In the present study, nine patients presented symptoms, of which seven were sensitivity disorders. Four of these were associated with type IV transmigration, with a close relationship with the inferior alveolar nerve.

Five patients presented additional impacted teeth, preferably with the inclusion of third molars (Table 1), although taking into account that the studied population is adult, this probability should be considered. Therefore, if the third molars are excluded, as Bertl et al. point out [11], the presence of other impacted teeth is not usually a common circumstance.

The development of tooth cysts on transmigrated canines can be explained by the degeneration of the follicular sac of the included tooth. Bertl et al. [11] and Sharma et al. [29] found percentages of 7.9% and 6.6%, respectively. In the present study, a total of six cases (11.53%) have been observed and confirmed by anatomopathological studies. These results agree with this obtained by Murat et al. [25], who found a total of 17 cysts (11.76%).

From a therapeutic point of view, the timing of diagnosis plays a crucial role in determining the treatment options and prognosis. These options include surgical treatment, autotransplantation, orthodontic treatment, or clinical and radiological monitoring [18,30].

When the diagnosis is early, surgical exposure followed by orthodontic treatment will always be the treatment of choice, so long as the transmigrated tooth’s inclination and position favor this option [4,5,9,31]. In the same line, Sinko et al. [31] emphasized the importance of patient compliance for all tooth-preserving treatment strategies (i.e., orthodontic alignment or autotransplantation). Autotransplantation can be a first-line tooth-preserving treatment strategy if the patient’s refused the orthodontic treatment. However, autotransplantation in transmigrated canines was poorly documented in the literature. Rafat and Ijaz [32] explained in their study that it is advisable to perform the procedure with care as it is technique-sensitive, and some of the factors responsible for the survival of the transplant included an open apical foramen and the continued vitality of the periodontal membrane. In cases where the periodontal ligament is traumatized during transplant, external root resorption or ankylosis is often observed. Kulkarni and Lee [33] presented a case of a 14-year-old patient with an apically closed, transmigrated permanent canine that was autotransplanted into its natural position without endodontic treatment and orthodontically aligned into the ideal occlusion. The transplanted canine maintained long-term tooth vitality, physiologic mobility, and normal masticatory function.

Nevertheless, in the present study, the most common treatment is surgical removal in 55.75% of patients. These findings would be explained for many reasons, including the older age of the patients who preferred a faster and simpler treatment.

It is important to note that in patients who opt to defer treatment and remain under observation, it is essential to maintain regular clinical and radiological check-ups and to warn them of possible complications derived from the impacted teeth, such as ankylosis, damage to adjacent structures, or the resorption of neighboring teeth.

## 5. Conclusions

Transmigrated canines are a rare entity and are usually diagnosed by chance after a routine radiographic examination. Given this finding, CBCT scanning makes diagnosis more precise in terms of the exact position of the canines and their relation to adjacent teeth and anatomical structures.

In most cases, transmigrated canines do not present symptoms; however, they could be associated with pathological processes such as cysts and tumors.

The therapeutic attitude contemplated is based on observation and follow-up, orthodontic treatment, and surgical removal. The age of diagnosis and the type of transmigration could be factors to be considered for repositioning in the arch with orthodontic treatment; surgical removal is the most common procedure, especially in cases in which transmigrated canines have associated pathologies, and the ‘wait and see’ response requires clinical and radiological follow-up to avoid future complications.

## Figures and Tables

**Figure 1 biology-11-01751-f001:**
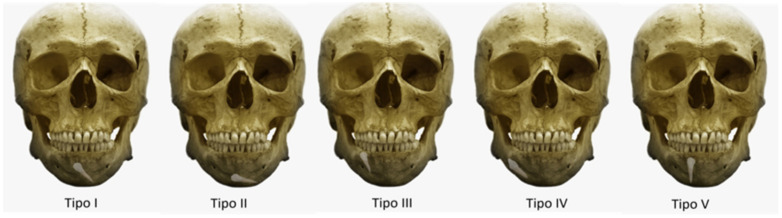
Mupparapu’s classification.

**Figure 2 biology-11-01751-f002:**
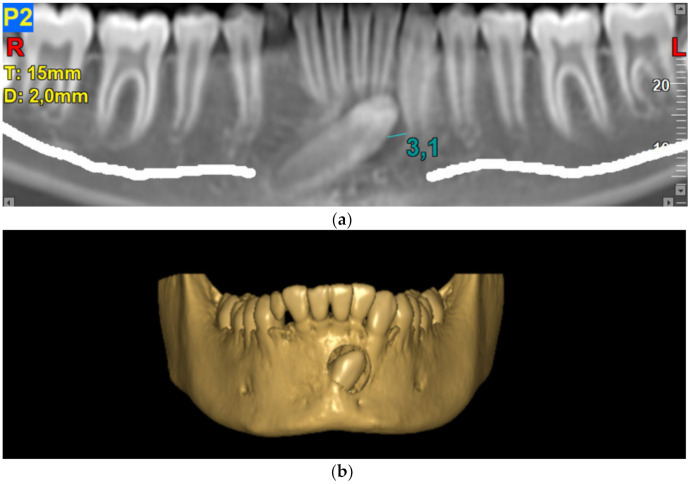
Transmigrated canine type I. (**a**) Panoramic view measuring follicular enlargement; (**b**) tridimensional view.

**Figure 3 biology-11-01751-f003:**
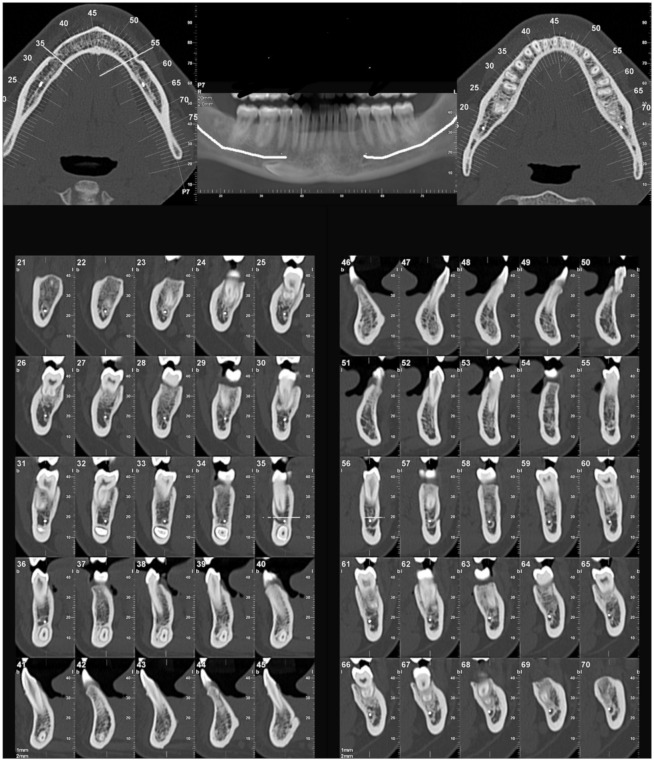
CBCT images with transmigrated canine type IV.

**Figure 4 biology-11-01751-f004:**
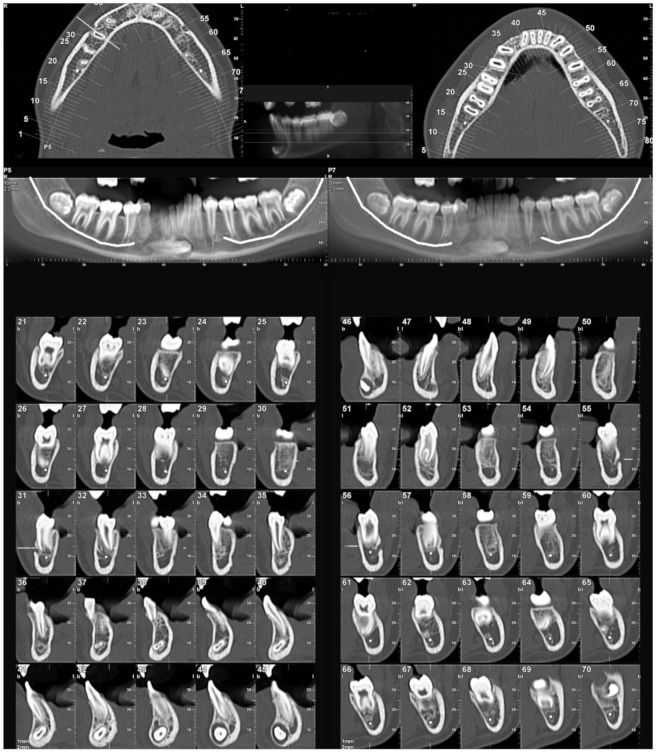
CBCT images with transmigrated canine type II.

**Figure 5 biology-11-01751-f005:**
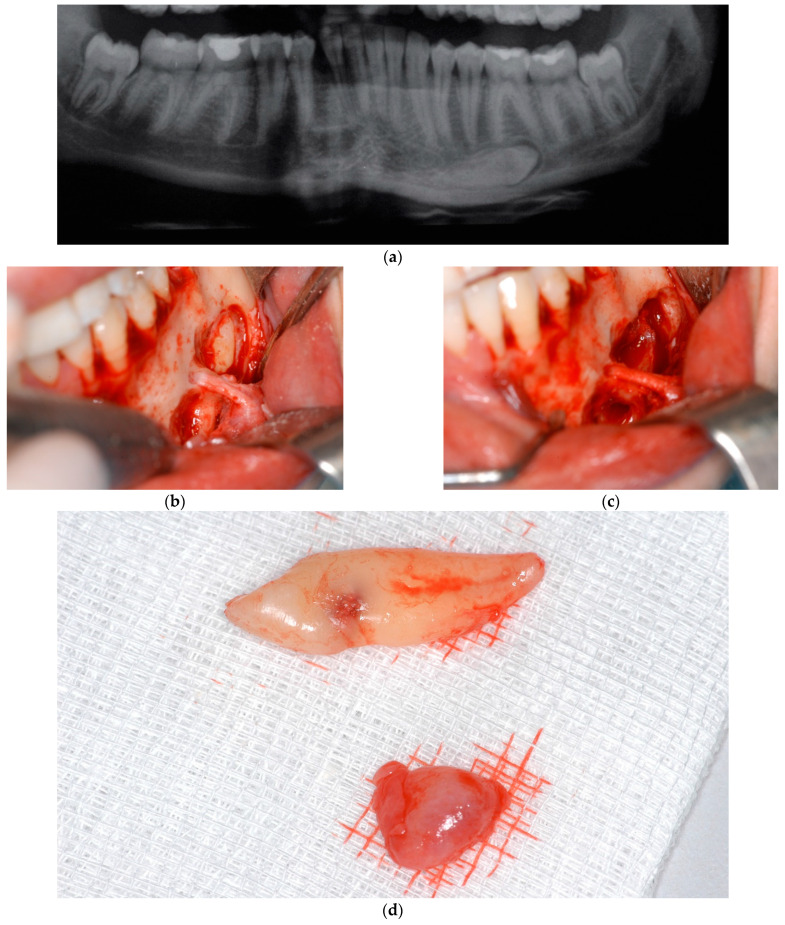
Transmigrated canine type IV. (**a**): Panoramic radiograph; (**b**): canine with a close relationship with mental nerve; (**c**): canine extraction with mental nerve preservation; (**d**): canine with the follicular sac.

**Table 1 biology-11-01751-t001:** Symptom Clinical and radiological data.

Transmigration Type *	Type I	Type II	Type IV	Type V	*p*=
	8	19	22	3	
Gender	4M/4F	10M/9F	9M/13F	1M/2F	0.911
Side	5L/3R	12L/7R	11L/11R	2L/1R	0.725
Symptoms	2	3	4	0	0.149
Other impactions and affected teeth	1	1	2	1	0.428
	18,28,13,23	18,28	18,38,48	18,28	
			28,38,48		
Follicular enlargement (means value)	2.75 mm	2.02 mm	2.06 mm	1 mm	0.245
Cysts	0	1	4	1	0.307

* Mupparapu´s classification. Gender: fale (M)/Female (F). Side: left (L)/right (R). Follicular enlargement: millimeters (mm).

## Data Availability

Databases used and/or analyzed during the current study are available from the corresponding author upon reasonable request.

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
