# Peer review of "Dental Transmigration: An Observational Retrospective Study OF52 Mandibular Canines"

_biology, 2022, doi:10.3390/biology11121751_

Round 1
Reviewer 1 Report
The article is not so interesting . The methodology isn't sound and the conclusions are a confirmation of known facts and there you don't mention all therapeutic options, like for example, autotransplantation.
Author Response
"Please see the attachment."

Reviewer 2 Report
Dear Authors,
thank you for submitting your paper. I found it very interesting, your population is very large, and represents a valuable tool. I suggest minor revision and a grammar check. Some parts of the paper are not well organised.
Author Response
"Please see the attachment."

Reviewer 3 Report
Dear Authors,
thank you for the opportunity to read your article. Although the study is not innovative, it provides valuable data on the occurrence of transmigration of lower canines in the population, therefore, in my opinion, it is worth publishing. Please make a few corrections to the manuscript:
- line 144: the skeletal class I was present in 49 patients,...class II in 2 patients,.. and class III in one patient,
- table 1: Symptoms (it is 'synthoms' in the table),
- line 210: Plakwicz et al.
Author Response
"Please see the attachment."

Round 2
Reviewer 1 Report
I 've read your corrections of english language and implementation of the contents of the discussion.